# Pseudoxanthoma Elasticum, Kidney Stones and Pyrophosphate: From a Rare Disease to Urolithiasis and Vascular Calcifications

**DOI:** 10.3390/ijms20246353

**Published:** 2019-12-17

**Authors:** Emmanuel Letavernier, Elise Bouderlique, Jeremy Zaworski, Ludovic Martin, Michel Daudon

**Affiliations:** 1UMR S 1155 and Physiology Unit, AP-HP, Hôpital Tenon, Sorbonne Université and INSERM, F-75020 Paris, France; eliseboud@aol.com (E.B.); jyzaworski@gmail.com (J.Z.); michel.daudon@aphp.fr (M.D.); 2PXE Consultation Center, MAGEC Reference Center for Rare Skin Diseases, Angers University Hospital, MITOVASC Institute, UMR CNRS 6015 INSERM U1083 Angers University, 49000 Angers, France; lumartin@chu-angers.fr

**Keywords:** pseudoxanthoma elasticum, pyrophosphate, kidney, Randall’s plaque

## Abstract

Pseudoxanthoma elasticum is a rare disease mainly due to *ABCC6* gene mutations and characterized by ectopic biomineralization and fragmentation of elastic fibers resulting in skin, cardiovascular and retinal calcifications. It has been recently described that pyrophosphate (a calcification inhibitor) deficiency could be the main cause of ectopic calcifications in this disease and in other genetic disorders associated to mutations of *ENPP1* or *CD73*. Patients affected by Pseudoxanthoma Elasticum seem also prone to develop kidney stones originating from papillary calcifications named Randall’s plaque, and to a lesser extent may be affected by nephrocalcinosis. In this narrative review, we summarize some recent discoveries relative to the pathophysiology of this mendelian disease responsible for both cardiovascular and renal papillary calcifications, and we discuss the potential implications of pyrophosphate deficiency as a promoter of vascular calcifications in kidney stone formers and in patients affected by chronic kidney disease.

## 1. Introduction: Pseudoxanthoma Elasticum and Related Diseases

### 1.1. Clinical Manifestations

Pseudoxanthoma elasticum (PXE; OMIM #264800, prevalence 1/25,000 to 1/50,000) is an autosomal recessive disease resulting mainly from mutations in *ABCC6* gene [1,2]. PXE is characterized by the fragmentation of elastin fibers, elastorrhexis, and calcifications of soft tissues involving mainly skin, arteries and retina [3,4]. The 2/3 female to 1/3 male ratio in patients diagnosed with PXE remains unexplained to date.

The disease may present during infancy but is classically discovered in teenagers or young adults, because of cutaneous manifestations [3,4,5,6].

Xanthoma-like papules may be sparse or form coalescent plaques and folds of esthetic concern. They give the name to the disease and predominate in flexion zones and neck. Skin biopsy and histopathological examination reveal calcifications with destruction of the elastic fibers [7]. It remains uncertain whether elastorrhexis precedes calcifications or whether calcified elastin fibers are broken in a second step.

PXE also affects retina, inducing fragmentation and rupture of calcified elastin fibers of the Bruch’s membrane. The examination of the fundus of the eye reveals a typical “peau d’orange” aspect in PXE patients, with angioid streaks and choroidal neovascularization leading to hemorrhages [8,9]. When affecting the macula, these lesions lead to the loss of central vision.

Cardiovascular calcifications are a hallmark of PXE but clinical manifestations are relatively delayed, predominating after the fourth decade of life, and their severity is extremely variable among PXE patients [5]. Peripheral arterial disease is frequent and there is a significantly increased risk of stroke [10,11]. Coronary calcifications are also frequent but the risk of cardiac infarction seems only mildly increased (if any) in comparison to the general population [5,12]. The earliest arterial calcifications observed in PXE affect elastic fibers of the medial layer, and predominate in medium and small-sized musculo-elastic arteries. Cardiovascular remodeling in large and medium sized musculo-elastic arteries is characterized by an increased intima-media thickening [13,14]. The arterial lesions affecting PXE peripheral arteries differ from those due to aging, hypertension or classic atherosclerosis and share similarities with calcifications and remodeling observed in chronic kidney disease [5,15].

To date, there is no specific curative or preventive therapy for PXE patients. Bisphosphonates such as Etidronate seem promising to prevent cardiovascular calcifications and anti-Vascular Endothelial Growth Factor (VEGF) intraocular injections limit neoangiogenesis [9,16,17].

Two other autosomal recessive mendelian diseases share phenotypic similarities with PXE: the generalized arterial calcifications of infancy (GACI), caused by mutations of the ectonucleotide pyrophosphatase phosphodiesterase NPP1 encoded by the *ENPP1* gene (OMIM 208000) and Arterial calcification due to deficiency of CD73 (ACDC, OMIM 211800), due to mutation of the *NT5E* gene encoding CD73, an ecto 5’-nucleotidase adenosine (ADO)-generating enzyme [18,19,20,21] GACI is an extremely severe but fortunately rare disorder characterized by extensive arterial calcification and stenosis affecting young children leading to heart failure. In some cases, the disease is less severe and patients with *ENPP1* mutations may present with the PXE phenotype. ACDC has been discovered more recently in a few families and is also characterized by cardiovascular calcifications and stenosis [19,20].

In addition to phenotypic resemblances, these three diseases share a common pathophysiological link: pyrophosphate (PPi) deficiency.

### 1.2. Pathophysiology of PXE, GACI and ACDC: Pyrophosphate Deficiency

During decades, the mechanism responsible for ectopic calcifications in PXE patients remained a mystery. *ABCC6* mutations are responsible for most of the PXE syndromes diagnosed. *ABCC6* encodes an adenosine triphosphate (ATP)-binding cassette transporter mainly expressed in the liver and to a lesser extent in kidney proximal tubular cells [1,22,23]. This specific expression raised the hypothesis that PXE would be a metabolic disease responsible for distant manifestations, i.e., connective tissue calcifications [24]. Due to its structure, close to other ABC-type transporters, ABCC6 could promote the efflux of calcification inhibitors from hepatocytes and tubular cells toward the systemic circulation [24].

A role for vitamin K has been suggested by several observations [25]. First, antivitamin K therapy is associated with vascular calcifications and sometimes calciphylaxis in hemodialyzed patients [26]. Second, *GGCX* mutations cause a PXE-like calcification phenotype and are associated with deficiency in vitamin K clotting factors [25]. The *GGCX* gene encodes a gamma-glutamyl carboxylase which catalyzes the conversion of glutamate residues to gamma-carboxyglutamate residues (gamma-carboxylation). This posttranslational modification process uses vitamin K as an essential cofactor. Gamma-carboxylation is necessary to activate multiple vitamin K-dependent proteins including coagulation factors (Factors II, VII, IX and X) and proteins such as Matrix Gla Protein (MGP). Third, PXE patients have lower serum levels of vitamin K in comparison with a control population [25]. Finally, MGP is a vitamin K-dependent mineralization inhibitor. *MGP* homozygous mutations are responsible for Keutel syndrome, characterized by cartilage calcification and pulmonary artery stenoses [27]. Nevertheless, administration of vitamin K was not efficient to reduce clinical manifestations in patients or to improve significantly the ectopic calcification which also occur in *Abcc6^−/−^* mice [28,29].

A major breakthrough in the field of calcifying diseases has been the identification that the hepatic (and probably tubular) ABCC6 transporter promotes the release of extracellular adenosine triphosphate (ATP), which serves as a substrate for NPP1 in the vasculature to generate adenosine monophosphate (AMP) and PPi, a potent anticalcifying molecule [30,31]. This process is the main source of circulating PPi, that counteracts the formation of calcium phosphate ectopic calcifications. PPi inhibits the crystallization and the growth of calcium phosphate crystalline phases such as hydroxyapatite [32]. Patients with PXE and GACI as well as *Abcc6^−/−^* and *Enpp1* mutant (tiptoe walking: ttw/ttw) mice have a reduced plasma PPi level, explaining, at least in part, their mineralization disorder [22]. Of note, PPi treatment is sufficient to prevent ectopic mineralization in both *Abcc6^−/−^* and *Enpp1^ttw/ttw^* murine models [33,34,35].

Moreover, ABCC6 and NPP1 combined activity generates AMP in addition to PPi. AMP is rapidly converted into adenosine by CD73, which exerts a tonic inhibition on tissue non-specific alkaline enzyme (TNAP). In ACDC, mutations in *NT5E/CD73* lead to insufficient adenosine levels, increasing TNAP activity: TNAP degrades PPi to generate inorganic phosphate and eventually promotes hydroxyapatite precipitation in ectopic tissues [22,33,35].

Thus, ABCC6, NPP1 and CD73 increase PPi systemic synthesis and decrease endogenous PPi degradation through TNAP inhibition (Figure 1).

## 2. Pseudoxanthoma Elasticum and Kidney Calcifications

### 2.1. Nephrocalcinosis and Kidney Stones: Preliminary Reports

A few preliminary and sparse reports have suggested that PXE patients may be affected by kidney stones [36]. In addition to kidney stones, the presence of typical nephrocalcinosis has been reported in a few patients affected by PXE [37,38,39]. When exposed to a procalcifying diet (high phosphate, vitamin D and low magnesium), *Abcc6*^−/−^ and *Enpp1^−/−^* mice develop rapidly a nephrocalcinosis consisting in multiple calcium phosphate tubular plugs [40].

### 2.2. High Prevalence of Kidney Stones in PXE Patients

In a recent PXE cohort study that was not specifically designed to record kidney stone prevalence, observations suggested that nephrolithiasis was an unrecognized but a prevalent feature of PXE, affecting at least 10% of patients [11]. To better characterize the prevalence of kidney stones among patients with PXE, we conducted a retrospective study in 170 patients participating in the Angers PXE cohort [41]. Among the 164 patients who received the survey, 113 fulfilled the questionaire (33 men and 80 women). Among these patients, 45 (39.8%) declared they had a past medical history of kidney stones (asymptomatic stones in four patients or renal colic in 41 patients). Although this type of study may be biased, as patients affected by kidney stones may be more prone to fulfill the survey, the prevalence of kidney stones and renal colic in PXE patients appears to be extremely high. As a matter of comparison, the lifetime expected incidence of kidney stones is nearing 10% in the general population. Usually, urolithiasis is more frequent in males than females but in PXE patients approximately two thirds of the patients are women. This is consistent with the higher prevalence of PXE in women.

The analysis of six computed tomography (CT)-scans from PXE patients revealed the presence of papillary calcifications, Randall’s plaques, the first step of kidney stone formation in five of them [41]. This is remarkable as only the presence of massive plaques can be determined by imaging. These calcifications affected rather the tip of the renal papilla than the renal medulla. One patient had both papillary calcifications and medullary calcifications, i.e., nephrocalcinosis. CT scans had been performed in patients with a past medical history of kidney stones, but one may hypothesize that patients with PXE and without a past medical history of urolithiasis may also be affected by asymptomatic papillary calcifications. We examined a limited number of CT-scans so that we cannot estimate the prevalence of papillary calcifications/Randall’s plaque and nephrocalcinosis in PXE patients. A few stones from PXE patients have been analyzed and presented Randall’s plaque fragments made of carbonated apatite, suggesting that stone formation in these individuals was initiated by these plaques. No specific study dedicated to the biological risk factors of kidney stone formation in PXE has been performed to date. In the same way, chronic kidney disease (CKD) is not a classic feature of PXE and we still ignore whether some PXE patients affected by kidney stones and papillary calcifications may be at risk to develop CKD.

### 2.3. Abcc6^−/−^ Mice: A Murine Model of Randall’s Plaque

*Abcc6^−/−^* mice recapitulate many of the phenotypical characteristics of the human PXE disease [42]. Mice are affected by ectopic calcifications involving aorta, heart, retina and vibrissae and their PPi serum level is lower than in control mice. As renal cortical and medullary calcifications had been described in these animals, especially after an “acceleration diet”, we hypothesized that papillary calcification could develop in this model [40]. Aging *Abcc6^−/−^* mice actually developed kidney calcifications electively at the tip of the papilla, mainly in the interstitial tissue, round shaped and surrounding the loops of Henle and the vasa recta: they differed from the tubular plugs observed after an acceleration diet. These calcifications were made of carbonated apatite spheres as evidenced by scanning electron microscopy and Fourier transform infrared microspectroscopy [41]. At the nanometer scale, by using transmission electron microscopy coupled to electron energy loss spectroscopy, we evidenced that incipient calcifications were made of concentric layers containing calcium and phosphate alternating with organic compounds, as described in incipient human calcifications forming the Randall’s plaque. Remarkably, the renal calcification observed in aging *Abcc6^−/−^* mice met the four essential criteria of Randall’s plaque: (i) interstitial deposits surrounding tubules and vasa recta, (ii) electively located at the tip of the renal papilla, (iii) made of carbonated apatite, and (iv) forming nanometer-scale elementary structures made of minerals and organic compounds.

*Abcc6^−/−^* mice had also a lower urinary excretion of PPi than control mice, and as previously described lower serum levels of PPi [41]. *Abcc6^−/−^* mice expressing a functional human ABCC6 protein in the liver had an increase in serum PPi circulating levels and were protected against kidney calcifications at 18 months of age [33]. It seems therefore very likely that the systemic PPi deficiency in these mice was at the origin of calcium phosphate supersaturation at the tip of the papilla, leading to papillary calcification.

## 3. The Mystery of Randall’s Plaque Formation: A Role for Calcification Inhibitors and Pyrophosphate?

### 3.1. Randall’s Plaque: the Origin of Renal Calculi

Randall’s plaque is the first step of calcium oxalate stone formation in many cases. Alexander Randall proposed its original theory in 1936 when he claimed the identification of the origin of renal calculi [43]. He performed extensive forensic studies and observed, at the tip of the renal papilla, whitish deposits made of calcium phosphate and carbonate, originating from the renal interstitium. In some cases, these lesions disrupted the urothelium cellular barrier and brown calcium oxalate stones were attached to these “Randall’s” plaques, forming a heterogeneous nucleation process. He identified that the calcium oxalate stones had an umbilication due the shape of the papilla and that calcium phosphate remnants, fragments of Randall’s plaque, were present in this umbilication once the stone was passed, identifying the origin of the stone. Other authors performed similar studies during this period [44]. It has been postulated early that the high concentration in calcium in renal papilla could be at the origin of the plaque. The interest in Randall’s plaque has grown during the last decades, as the development of endourology evidenced the presence of stones attached to Randall’s plaque in situ. Randall’s plaque was found in stone formers in 57%–99% of patients when reno-ureterocopy was performed. The prevalence seems lower in France than in North American studies [45,46,47,48,49]. Although Randall’s plaques seem to be very frequent, common papillary calcifications are usually too small to be evidenced by CT-scan (unlike PXE patients).

### 3.2. Determinants of Randall’s Plaque Formation

The main risk factors responsible for calcium oxalate supersaturation and crystals or stone formation are relatively well identified: low diuresis, hypercalciuria, hyperoxaluria and/or hypocitraturia [47,50]. However, the origin of the Randall’s plaque itself is less understood. The prevalence of calcium oxalate stones is increasing worldwide [47,51]. As there is no longitudinal study taking into account stone morphology or papillary plaque coverage, the role of Randall’s plaque in this epidemic is difficult to assess. However, we identified that the proportion of stones with a papillary umbilication and typical plaque remnants was three times more frequent in recent years than three decades ago, especially in young adults [52]. In 2003, Evan et al. analyzed papilla biopsies and identified that incipient Randall’s plaque at the tip of the renal papillae were made of subepithelial apatite deposits surrounding the loop of Henle [53]. These observations suggested that apatite droplets formed in around the loop of Henle in contact with basement membrane could spread in the interstitium and form plaques. It has been suggested that calcium phosphate supersaturation in the thin limb of the loop of Henle could promote calcium phosphate particle precipitation and that some of these particles attached to epithelial cells could migrate by endocytosis to the interstitial tissue and form plaques [54,55]. We performed analyses of incipient Randall’s plaque in dozens of healthy papillae and identified that early calcifications could also appear around vasa recta at the tip of renal papilla, as previously highlighted by Stoller et al. [56,57]. This suggests that calcium phosphate supersaturation should be extremely high in this environment. Kuo et al. have identified that hypercalciuria, low diuresis and low urine pH correlated with the surface of Randall’s plaque in kidney stone formers [58]. More than seven decades ago, Vermooten had performed forensic studies in South Africa and identified Randall’s plaque in 17% of whites and only in 4% of the local native Bantus, known to have low calciuria and to be rarely affected by calcium-related stones [44,59]. Hypercalciuria could therefore be involved first in Randall’s plaque formation and several years or decades later in calcium oxalate stone formation, once plaque has developed and eroded urothelium locally.

### 3.3. Randall’s Plaque and Calcification Inhibitors

Calcium concentration is predicted to be extremely high at the tip of the renal papilla, especially around vasa recta [60,61]. Considering (i) that phosphate concentration is certainly significant, although reabsorbed partly in the proximal tubule, and (ii) that papillary pH should stand within physiological range (7.4), calcium phosphate supersaturation should be extremely high at the tip of the papilla. This suggests that calcification inhibitors play a key role to prevent Randall’s plaque formation. A very large number of low-weight and macromolecular calcification inhibitors have been studied in the field of urolithiasis [62]. Currently, there is no monogenic mendelian disease responsible for calcification inhibitor deficiency and kidney stone formation, maybe with the exception of distal renal tubular acidosis, which is responsible for hypocitraturia and nephrocalcinosis [63]. Interestingly, *ABCC6* mutation in PXE patients and *Abcc6* knock-out murine models are sufficient to induce papillary calcification [41]. This monogenic defect responsible for PPi deficiency is clearly not compensated by other calcification inhibitors. This observation is remarkable as the development of kidney tissue calcifications is extremely difficult to achieve in rodent models. We previously tried to induce Randall’s plaque in Sprague Dawley rats exposed to vitamin D and calcium supplementation for 6 months. These animals developed tubular plugs and stones but none of them developed papillary interstitial calcifications [64]. Similar observations have been performed in the genetic hypercalciuric stone forming rat extensively studied by Bushinsky et al. [65]. These data confirm that calcification inhibitors, especially PPi, play a major role to prevent Randall’s plaque formation.

Further studies are required to assess whether kidney stone formers affected by Randall’s plaque have lower urine (or serum) PPi levels than kidney stone formers without Randall’s plaque or than general population. Of note, seminal studies had previously highlighted that kidney stone formers may have lower urinary PPi excretion than matched subjects [66]. However, this field of research has probably been limited during the last decades by the lack of reliable methods to measure PPi, due to pre-analytic pitfalls and to the very low concentration of PPi in biological fluids.

### 3.4. The Abcc6^−/−^ Murine Model: A Tool to Identify Randall’s Plaque Determinants and New Therapeutics

*Abcc6^−/−^* mice expressing a functional human ABCC6 protein in the liver or supplemented with PPi were protected against kidney calcification [33,34]. It has been shown that oral PPi is partly absorbed in human subjects and increases serum PPi circulating levels. Taken together, these data suggest that PPi or drugs increasing PPi synthesis could potentially be used as a treatment against kidney calcifications and Randall’s plaque.

*Abcc6^−/−^* mice can also be used to test whether drugs or dietary factors could influence Randall’s plaque formation. We hypothesized that the increased proportion of stones grown on Randall’s plaque during the past decades could, at least partly, be explained by the widespread use of vitamin D supplementation in the general population [52]. Although the question is extremely controversial, some studies highlighted a relationship between vitamin D intakes, especially in addition with calcium supplementation, and stone formation [67]. *Abcc6^−/−^* and wild-type mice were exposed to vitamin D supplementation, with or without a calcium-rich diet, to promote the formation of Randall’s plaque. The combined administration of vitamin D and calcium accelerated significantly Randall’s plaque formation in *Abcc6^−/−^* mice, although the animal did not develop hypercalcemia or even a significant increase in urine calcium excretion [68]. This model raises some concerns regarding the potential role of long-term vitamin D supplementation and calcium intakes in predisposed individuals.

## 4. Pyrophosphate Deficiency, the Common Link between Vascular Calcifications, Kidney Stones and Chronic Kidney Disease?

Epidemiological studies have highlighted an association between nephrolithiasis and systemic conditions including cardiovascular diseases but the underlying mechanisms remain unidentified [69]. Vascular calcifications are highly associated with cardiovascular mortality. Kidney stone formers, even young patients, have an increased carotid intima-media wall thickness and an increased arterial stiffness in comparison with matched populations [70,71]. Kidney stone formers have a higher degree of aortic calcification than age- and sex-matched non-stone formers and more severe coronary artery calcification, suggesting that vascular calcifications and kidney stones may have a common underlying mechanism [72].

The potential implication of PPi in vascular calcification is not a novel hypothesis, as Schibler et al. have reported in 1968 that PPi and polyphosphate can inhibit the induction of aortic calcifications by vitamin D, but the recent evidence that PXE patients are affected by both cardiovascular calcifications and kidney stones and Randall’s plaque suggests that PPi deficiency could be involved in the development of vascular calcifications in kidney stone formers [41,73]. Here again, further studies based upon a reliable assay of serum PPi levels would be necessary to assess whether PPi deficiency is the missing link between kidney stone/Randall’s plaque and vascular calcifications.

Moreover, a relative deficiency in PPi could be involved in the vascular calcifications observed in patients affected by CKD, responsible for an increased cardiovascular morbidity and mortality [74]. TNAP activity is increased in uremic rats, reducing PPi/inorganic phosphate ratio and ABCC6 expression is reduced in uremic mice and rats [75,76]. Exogenous PPi could in theory be of help to protect against vascular calcifications in CKD.

## 5. Conclusions

An increased prevalence of kidney stones has been reported recently in patients affected by PXE, a disease characterized by low circulating PPi levels and vascular calcifications. These patients are affected by massive Randall’s plaques and in some cases by nephrocalcinosis. In clinical practice, patients affected by PXE should be aware that they are at high risk to form kidney stones, and preventive measures should be considered, as for any kidney stone former (increased diuresis, decreased urine calcium excretion, etc.). These observations from a rare monogenic disease evidence that PPi probably prevents the development of Randall’s plaque and, in the long term, the formation of kidney stones, a disease with a lifetime incidence nearing 10% of the population. Further studies are required to assess whether PPi deficiency is involved in Randall’s plaque and kidney stone formation in the general population, but also whether administration of PPi could be protective in that setting. *Abcc6^−/−^* mice can also be used as a model of Randall’s plaque to test new drugs or identify potential determinants of plaque formation or worsening. In addition, PPi could be a missing link between kidney stones and the high prevalence of cardiovascular calcifications observed in kidney stone formers, a deficiency in PPi and potentially in other calcification inhibitors could promote both papillary and vascular mineralization.

## Figures and Tables

**Figure 1 ijms-20-06353-f001:**
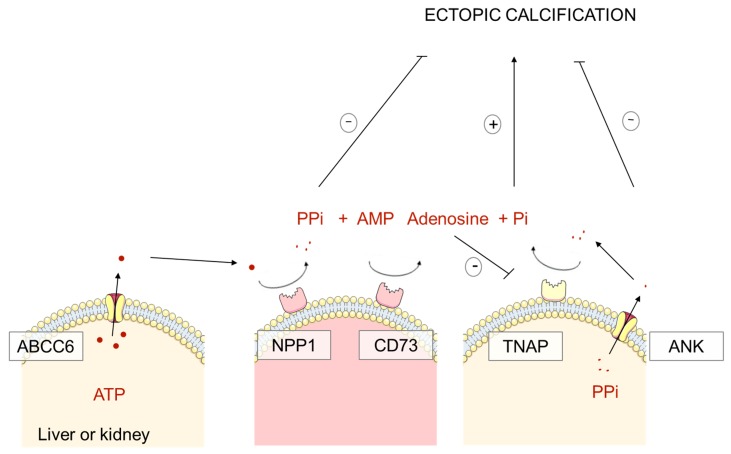
Systemic Pyrophosphate (PPi) synthesis. ABCC6 (expressed in hepatocytes and proximal tubular cells) and NPP1 (in arteries and capillaries) combined activity generates AMP in addition to PPi. AMP is rapidly converted into adenosine by CD73, which exerts a tonic inhibition on tissue non-specific alkaline enzyme (TNAP). TNAP degrades PPi to generate inorganic phosphate and promotes hydroxyapatite precipitation in ectopic tissues. ANK allows PPi externalization from cells, its role in ectopic calcifications is unclear.

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
