# Peer review of "Pseudoxanthoma Elasticum, Kidney Stones and Pyrophosphate: From a Rare Disease to Urolithiasis and Vascular Calcifications"

_ijms, 2019, doi:10.3390/ijms20246353_

Round 1

Reviewer 1 Report

“GGCX mutations”: please spell out and explain what this gene codes for. “Vitamin K clotting factors”: which “vitamin K-dependent clotting factors”? (as you call MGP) referring to your PXE cohort you say “Although this type of study may be biased”: what bias are you describing or suggesting? “The analysis of several computed tomography (CT)-scans from PXE patients revealed the 139 presence of papillary calcifications, Randall’s plaques”: many authors state that Randall plaques are NOT seen on CT scan. Are you confident you are describing Randall plaques and not nephrocalcinosis? In fact you then say “only the presence of massive plaques can be determined by imaging”. You also then refer to “massive Randall’s plaque fragments”: I am not sure there is such a thing. You do not state what the composition of these “massive Randall plaques is”: one would expect calcium phosphate, either hydroxyapatite or brushite, as you state in the next paragraph. Suggest you change “Caucasian” to white, to get away from that ancient and inaccurate racial classification. In describing the phenotype of PXE mutations you do not state whether CKD is a feature, though you imply this later when you discuss the relationships of calcification and stones; if so, related to stones or to nephrocalcinosis or both? Is there a useful way to supplement PPi orally? Is that possible?

Author Response

Thanks for reviewing and positive comments

“GGCX mutations”: please spell out and explain what this gene codes for. “Vitamin K clotting factors”: which “vitamin K-dependent clotting factors”? (as you call MGP)

Reply: GGCX encodes a gamma-glutamyl carboxylase which catalyzes the conversion of specific glutamate residues to gamma-carboxyglutamate residues, a process called gamma-carboxylation. This posttranslational modification process uses vitamin K as an essential cofactor. Gamma-carboxylation is essential in the activation and proper functioning of multiple VK-dependent proteins, the most well-known of which are involved in blood clotting, including coagulation factors (FII, FVII, FIX and FX) and natural anti-clotting agents involved in soft tissue mineralization such as matrix gla protein (MGP).

We clarified this chapter in the revised version of the manuscript

referring to your PXE cohort you say “Although this type of study may be biased”: what bias are you describing or suggesting? “

Reply: One may hypothesize that patients who experienced renal colic may be more prone to fulfill the survey, increasing the number of cases. Nevertheless, even if considering that all patients who did not reply to the survey were not affected by kidney stones, the proportion of kidney stones would still remain extremely high (25%). We explain this bias in the revised version of the manuscript.

The analysis of several computed tomography (CT)-scans from PXE patients revealed the presence of papillary calcifications, Randall’s plaques”: many authors state that Randall plaques are NOT seen on CT scan. Are you confident you are describing Randall plaques and not nephrocalcinosis? In fact you then say “only the presence of massive plaques can be determined by imaging”.

Reply: The CT-scan images were kindly reviewed by Pr Marsault, a trained radiologist, who confirmed that these were papillary calcifications rather than stones. Most of Randall’s plaques are too small to be seen on CT-scan but RPs are observed more and more frequently with modern CT-scans. Data are still sparse in the literature but ongoing studies (in the Mayo Clinic for instance) will certainly provide interesting  data in a next future. These papillary calcifications are electively located at the tip of the renal papilla and differ from classic “nephrocalcinosis” which affects medulla and sometimes kidney cortex. One patient had both papillary calcifications and nephrocalcinosis. We tried to clarify the chapter in the revised version of the manuscript

You also then refer to “massive Randall’s plaque fragments”: I am not sure there is such a thing. You do not state what the composition of these “massive Randall plaques is”: one would expect calcium phosphate, either hydroxyapatite or brushite, as you state in the next paragraph

Reply: Randall’s plaque fragments were seen on stones that passed spontaneously. We did not collect a significant amount of stones to draw any definitive conclusion but in stones analyzed the fragments of Randall’s plaque were predominantly made of apatite (no brushite), as now mentioned.We removed the term “massive” as stones were relatively small but the plaque remnants were evident (not small apatite deposit in the papilla umbilication as frequently seen).

Suggest you change “Caucasian” to white, to get away from that ancient and inaccurate racial classification.

Reply: Actually the term was used in 1942. The change has been done

In describing the phenotype of PXE mutations you do not state whether CKD is a feature, though you imply this later when you discuss the relationships of calcification and stones; if so, related to stones or to nephrocalcinosis or both?

Reply: CKD does not seem a frequent feature. Renal function (according to serum creatinine) seems normal in a majority of PXE patients in our experience (more than 170 patients). We cannot exclude that individuals affected by nephrocalcinosis may develop CKD at term but we don’t have these data yet. 

We discuss this point briefly

 Is there a useful way to supplement PPi orally? Is that possible?

Reply: Oral PPi intakes increase serum PPi in normal subjects, as recently described by A. Varadi and his group. Experimental studies previously published (and ongoing) evidenced that oral PPi may reduce calcifications, in Abcc6 KO mice for instance. Further studies will tell us whether this approach may be useful in clinical practice. This possibility is just mentioned in the conclusion as it remains speculative to date.

Reviewer 2 Report

Thank you for this interesting and well written review about the possible connection between etiological factors of calcification of soft tissues ant papillary kidney stones, based on pseudoxanthoma elasticum characteristics.

I have a few comments:

- Section 2.2 (lines 139-140). It would be clarifying for the reader that the authors indicate the percentage of PXE patients presenting papillary calcifications, and compare that percentage with the corresponding to stone formers. Indeed, in section 3.1 (line 93) it is reported that a 57-99% of stone formers presented Randall’s plaque when reno-ureteroscopy was performed. This aspect should be clarified.

- Section 2.2 (lines 146-149). The authors refer to the results of an unpublished study. Since data are not available in the literature, the authors are requested to include such data, since it would again be clarifying for the reader to dispose of the values, specifically the serum and urine concentrations of pyrophosphate and the comparison with normal values.

Author Response

Thank you for this interesting and well written review about the possible connection between etiological factors of calcification of soft tissues ant papillary kidney stones, based on pseudoxanthoma elasticum characteristics.

Reply: thanks for the positive comments

I have a few comments:

- Section 2.2 (lines 139-140). It would be clarifying for the reader that the authors indicate the percentage of PXE patients presenting papillary calcifications, and compare that percentage with the corresponding to stone formers. Indeed, in section 3.1 (line 93) it is reported that a 57-99% of stone formers presented Randall’s plaque when reno-ureteroscopy was performed. This aspect should be clarified.

Reply: We tried to clarify this part of the manuscript. Actually, we still ignore the percentage of PXE patients affected by RP. In kidney stone formers, the evaluation was performed by ureteroscopy but in PXE patients we had only a few CT-scans from patients previously affected by kidney stones. It seems difficult to make a comparison but the interesting point is that RPs are extremely developed in PXE patients as they are seen on CT-scan

- Section 2.2 (lines 146-149). The authors refer to the results of an unpublished study. Since data are not available in the literature, the authors are requested to include such data, since it would again be clarifying for the reader to dispose of the values, specifically the serum and urine concentrations of pyrophosphate and the comparison with normal values.

Reply: We suppressed this paragraph as our colleagues in Angers who take care of PXE patients prepare original manuscripts including serum and urine biochemistry from PXE patients, with techniques recently upgraded to obtained reliable measurement of PPi in biological fluids: it may be a problem if we publish original data too early. Thanks for your comprehension.